# Valproic Acid Inhibits Progressive Hereditary Hearing Loss in a KCNQ4 Variant Model through HDAC1 Suppression

**DOI:** 10.3390/ijms24065695

**Published:** 2023-03-16

**Authors:** Yoon Seok Nam, Young Mi Choi, Sungsu Lee, Hyong-Ho Cho

**Affiliations:** Department of Otolaryngology-Head and Neck Surgery, Chonnam National University Medical School and Chonnam National University Hospital, Gwangju 61469, Republic of Korea

**Keywords:** KCNQ4, valproic acid, hearing loss, HSP90β, HDAC inhibitor, HDAC1

## Abstract

Genetic or congenital hearing loss still has no definitive cure. Among genes related to genetic hearing loss, the potassium voltage-gated channel subfamily Q member 4 (KCNQ4) is known to play an essential role in maintaining ion homeostasis and regulating hair cell membrane potential. Variants of the KCNQ4 show reductions in the potassium channel activity and were responsible for non-syndromic progressive hearing loss. KCNQ4 has been known to possess a diverse variant. Among those variants, the KCNQ4 p.W276S variant produced greater hair cell loss related to an absence of potassium recycling. Valproic acid (VPA) is an important and commonly used histone deacetylase (HDAC) inhibitor for class I (HDAC1, 2, 3, and 8) and class IIa (HDAC4, 5, 7, and 9). In the current study, systemic injections of VPA attenuated hearing loss and protected the cochlear hair cells from cell death in the KCNQ4 p.W276S mouse model. VPA activated its known downstream target, the survival motor neuron gene, and increased acetylation of histone H4 in the cochlea, demonstrating that VPA treatment directly affects the cochlea. In addition, treatment with VPA increased the KCNQ4 binding with HSP90β by inhibiting HDAC1 activation in HEI-OC1 in an in vitro study. VPA is a candidate drug for inhibiting late-onset progressive hereditary hearing loss from the KCNQ4 p.W276S variant.

## 1. Introduction

Currently, the genetic cause of hearing loss (HL) is being extensively investigated. It has been reported that genetic causality is detected in 50% of early-onset deafness [1]. Nonetheless, there are currently no definitive methods with which to cure congenital or genetic HL. Hearing aids and cochlear implantations are still the main rehabilitation methods for congenital HL [2]. Identifying the problematic gene is more for predicting the prognosis and subsequent patient counseling. Late-onset HL acquires hearing problems after birth. In contrast to congenital HL, late-onset HL provides a larger period for interventions, such as gene editing or gene replacement. Clinical trials for HL gene therapy have begun for gene defects such as GJB2 or Otoferlin (OTO-825 and AK-OTOF, respectively). The slower the progression of hearing loss, the larger the time period will be for the intervention.

The potassium voltage-gated channel subfamily Q member 4 (KCNQ4) is highly expressed in the outer hair cells of the organ of Corti [3]. It is also expressed in the brainstem, especially along the central auditory pathway [4]. In the outer hair cell, the KCNQ4 is expressed in both the basal and lateral membranes immediately following birth. However, it is expressed only in the basal membrane after the onset of hearing (P14 in mice) [4,5]. The expression is higher in the basal turn and weaker in the apical turn. It was suggested that the KCNQ4 extrudes K+ ions that entered outer hair cells from the apical membrane [6]. KCNQ4 mutations are etiologically linked to one of the most common types of non-syndromic HL: deafness non-syndromic autosomal dominant 2 (DFNA2) [7,8,9]. Fortunately, hearing loss in DFNA2 patients exhibits a late onset and progresses slowly over decades [8]. KCNQ4 knock-out mice demonstrate a similar pattern, thus providing a good model for late-onset progressive hereditary HL. Using these mouse models, it was suggested that DFNA2 HL is caused by the slow progressive damage of the outer hair cells through chronic depolarization [6].

Histone deacetylase (HDAC) is an important player in chromatic structure condensation and suppressing gene expression [10]. HDAC inhibitors have been used for a wide variety of purposes including anticancer, anti-aging, anti-inflammatory, antioxidative, and neuroprotection [11]. Similarly, the ability of HDAC inhibitors, SAHA and trichostatin A, to manage hearing deficits including drug-induced or noise-induced HL has been investigated [11]. Treatment with sodium butyrate showed a protective effect on gentamicin-induced hair cell loss through HDAC1 modulation [12]. Valproic acid (VPA), originally an anticonvulsant, also provided an HDAC inhibitory effect and regulated HDAC class I and II [13]. Interestingly, VPA showed its antiepileptic effect by preserving the KCNQ family M-current activity [14]. In addition, the combination of CHIR99021 and VPA is in the clinical trials (FX-322) to treat sensorineural hearing loss [15]. HDAC inhibitors, which are central players in epigenetic gene modification and regulation of intracellular signaling, are involved in HL gene regulation [11]. Therefore, VPA treatment is also well suited for modulating HL in a KCNQ4 variant model.

Herein, we investigated the protective effects of VPA on the auditory functions of the KCNQ4 variant and analyzed the feasibility of using an HDAC inhibitor to modulate late-onset genetic HL.

## 2. Results

### 2.1. Valproic Acid (VPA) Inhibits Progressive Hearing Deterioration in a KCNQ4 p.W276S Variant Mouse Model

Kharkovets et al. [6] reported that staining for the KCNQ4 protein revealed its presence at the base of the wild-type (WT) outer hair cell (OHC). Likewise, the KCNQ4 was detected in the OHCs from WT mice (Appendix A). The KCNQ4 p.W276S variant in 4–16-week-old mice showed significantly reduced hearing by click ABR (Appendix A). In addition, there was also a significantly decreased tone burst ABR in the KCNQ4 p.W276S variant mice (Appendix A). The KCNQ4 p.W276S variant led to the onset of deafness in mice before 4 weeks. Therefore, VPA (200 mg/kg body weight, once daily) treatment was administered by intraperitoneal (IP) injection to wild-type (WT, +/+), heterozygote (hetero, KI/+), and homozygote (homo, KI/KI) KCNQ4 p.W276S variant mice from 3 weeks to 6 weeks (21 days) (Figure 1A). At an age of 6 weeks, the hearing was normal in WT mice (30 ± 0 dB SPL), whereas it worsened in the hetero mice (75 ± 5 dB SPL). While the homo mice were nearly deaf (92.6 ± 1.94 dB SPL) without the VPA injection, the administration of VPA significantly improved the hearing of the homo mice (83 ± 1.72 dB SPL). Moreover, VPA treatment had a similar tendency in the hetero mice, although it was not statistically significant (70 ± 4.53 dB SPL). The VPA injection to WT mice produced no harmful effects and showed similar results to the non-treated WT (30 ± 0 dB SPL) (Figure 1B). Furthermore, tone burst ABR significantly attenuated the HL at 8, 16, 24, and 32 kHz in the homo KCNQ4 p.W276S variant mice (Figure 1C). The continuous administration of VPA by osmotic pump (200 mg/kg, from 3–7 weeks) also presented significant hearing preservation at the 5- and 6-week timepoints (Appendix A). Ultimately, all the treatment groups lost their hearing at 7 weeks. This result demonstrates that VPA can preserve the auditory function and delay the onset of HL in a KCNQ4 p.W276S variant in vivo.

### 2.2. Systemic Injection of VPA Directly Affects the Cochlea

We examined the gene expression in the brain and cochlear following the IP injection to determine whether the protection provided by VPA was due to a direct effect on the cochlea and brain. An intravenous injection of VPA (200 mg/kg) was administered, after which VPA activated the survival motor neuron (SMN) gene mRNA expression [16]. The expression of the SMN gene was increased in the brain of VPA-treated mice compared to the non-treated control brains (3.1 ± 0.4 vs. 1 ± 0.1) (Figure 2A). In the cochlea, the expression of SMN was 2-fold higher in the VPA-treated group than the non-treated group (2 ± 0.4 vs. 1 ± 0.2) (Figure 2B). According to Lauren E et al. [17], VPA treatment is associated with histone acetylation levels and SMN gene expression through HDAC2 inhibition. To determine if HDAC is inhibited by VPA treatment in the cochlea, we stained for histone H4 acetylation and analyzed by immunostaining. In all three parts of the cochlea (apex, middle, and base), the histone acetylation staining was more pronounced in the SV (stria vascularis), SL (spiral ligament), OC (Organ of Corti), and SG (spiral ganglion) (Figure 2C, white square) of the VPA-treated cochlea when compared to the VPA untreated mice (Figure 2C), as evidenced by the histology analysis (Figure 2D–F). Therefore, we concluded that a VPA injection can provide direct effects inside the cochlea.

### 2.3. VPA Rescues the OHC from Cell Death in the KCNQ4 p.W276S Variant

We showed that the KCNQ4 variant-induced hearing loss is improved by VPA treatment, and the cochlea were investigated using whole-mount immunofluorescence (Figure 3A). The OHC cell staining was conducted using Prestin and revealed that the KCNQ4 homo p.W276S variant contained a large number of OHC loss in the middle and base areas (Figure 3B). Staining with Myo7a showed that there was no damage to the IHCs (Figure 3B). The number of OHCs was decreased at all levels of the cochlea in the VPA non-treated homo group, although predominantly at the base turn. However, the number of OHCs was higher in the VPA-treated homo mice at the base and the rest of the cochlea turns. Despite a substantial loss of OHCs in the KCNQ4 hetero and homo p.W276S variants, in the middle and the base, the VPA treatment significantly improved the overall OHC survival (Figure 3B). Thus, by reducing HDAC activity with VPA, the OHCs were protected from cell death in the KCNQ4 p.W276S variant mice.

### 2.4. VPA Upregulated KCNQ4 Expression by Inhibiting HDAC1 Activity

Next, we demonstrated that VPA regulated the expression of KCNQ4 and HSP90β in HEK293T cells. VPA treatment promoted the expression of KCNQ4 in *pcDNA3.1-KCNQ4-EGFP-8xHis* transfected HEK293T cells (Figure 4A). Similarly, the expression of transfected *pcDNA3-HA-HSP90β* was increased using VPA treatment (Figure 4B).

VPA is known to be an HDAC class I and class IIa specific inhibiter [18]. We performed several class I and class IIa HDAC overexpression tests in HEK293T cells to determine which HDAC member was regulated by the VPA treatment in the cochlea (Appendix A). Indeed, transfection of *pCs2+-3myc-HDAC1* led to overexpression in HEK293T cell lines, which reduced the expressions of KCNQ4 and HSP90β (Appendix A). Likewise, transfection of *pCs2+-3myc-HDAC1* decreased the expressions of *pcDNA3.1-KCNQ4-EGFP-8xHis* and *pcDNA3-HA-HSP90β* in a dose-dependent manner (Figure 4C,D). In addition, endogenous KCNQ4 and HSP90β expressions were reduced by the overexpression of HDAC1 in a dose-dependent manner (Figure 4E).

### 2.5. VPA-Induced HSP90β Expression and HSP90β–KCNQ4 Interaction

Yanhong et al. [19] reported that HSP90α and HSP90β possess key roles in controlling the KCNQ4 homeostasis through the HSP40-HSP70-HOP-HSP90 chaperone pathway and the ubiquitin-proteasome pathway. Indeed, HSP90α and HSP90β both bind to KCNQ4 and exert opposite effects on the KCNQ4 channel. Most importantly, HSP90β restored KCNQ4 surface expression in cells mimicking the heterozygous conditions of DFNA2 patients.

Next, we performed an immunoprecipitation assay and transfected HEK293T cells with the *pcDNA3.1-KCNQ4-EGFP-8xHis* and *pcDNA3-HA-HSP90β* plasmid. An anti-GFP antibody was used for the KCNQ4 immunoprecipitation, and HSP90β was detected with an anti-HA (HSP90β) antibody. The KCNQ4 successfully recruited HSP90β and the VPA treatment increased binding in HEK293T cells (Figure 5A). However, HDAC1 (myc) interrupted the HSP90β (HA)–KCNQ4 (His) binding by recruiting KCNQ4 (Figure 5B). This result showed that KCNQ4 binding competed with HDAC1 and HSP90β and that the KCNQ4–HDAC1 binding affinity was stronger than that of KCNQ4–HSP90β. Overall, VPA treatment upregulated HSP90β and increased HSP90β–KCNQ4 binding by inhibiting HDAC1 activation. Altogether, these results suggest that VPA treatment represents an intriguing candidate for the future therapy of DFNA2 patients (Figure 6).

## 3. Discussion

This study is the first to elucidate that hearing loss induced by the KCNQ4 variant can be protected by inhibiting HDAC1 activation. In addition, we also showed that HDAC1 can regulate KCNQ4 and HSP90β expression through its interaction with KCNQ4. The overexpression of HDAC1 downregulated KCNQ4 and HSP90β expression and disrupted the binding of KCNQ4 to HSP90β. Thus, we propose that HDAC1 is an upstream signal transduction regulator of KCNQ4 and HSP90β. 

VPA activated its known downstream target, the SMN gene within the cochlea [17]. VPA treatment in mouse brain tissues increased histone acetylation levels, while associated HDAC2 levels increased at the SMN transcriptional start site. It represents an effective strategy for the treatment of spinal muscular atrophy (SMA). Brain and cochlea SMN mRNA expressions were markedly increased by VPA treatment (Figure 2A,B), while VPA treatment also increased histone H4 acetylation in the cochlea (Figure 2C). Thus, the intraperitoneal injection of VPA produced notable effects on the cochlea apex, middle, and base of the SV, SL, OC, and SG signaling regulations (Figure 2D–F). Our data suggest that VPA can reach the cochlea and works directly inside the cochlear tissue, instead of causing a secondary effect through other organs. 

VPA is known to be the elevated level of gamma-aminobutyric acid (GABA) in the brain. It modulates neuronal discharge by increasing GABA levels in pre- and postsynaptic neurons. Therefore, VPA enhances GABA activation and amplifies the neuronal response [20]. In the cardiovascular system, VPA ameliorated cardiac dysfunction, cardiac hypertrophy, and fibrosis [21]. Furthermore, VPA improves glycemic control by enhancing the number of β-cells in rat diabetic models [22]. In a model of diabetic nephropathy, Sun et al. found that VPA attenuated diabetes-mediated renal injury by suppressing endoplasmic reticulum-induced stress and apoptosis [23]. In addition, VPA also showed anticancer actions, whereby MCF-7 breast carcinoma cells, NCI-1299 and NCI-H460 lung cancer cells, NB-2 and UKF-NB-3 neuroblastoma cell lines, and HL-60 leukemia cells all underwent enhanced VPA-mediated apoptosis [24]. 

Previous studies indicated that HDACs are associated with the development and progression of hearing loss. The HDAC inhibitor, TSA, showed a protective effect on gentamicin-induced hair cell loss [25]. Moreover, pretreatment with SAHA markedly reduced noise-induced OHC loss in the NIHL model [26]. Likewise, SAHA could protect against cisplatin ototoxicity [27]. Furthermore, the HDAC inhibitor, sodium butyrate, reduced hair cell loss [12]. Recent studies have elucidated that HDAC functions are implicated in hair cell death characterized by inflammation and ROS production. VPA treatment caused an increase in histone H4 acetylation in the SV, SL, OC, and SG regions. These suggest that HDAC activity is involved in the development of hearing loss and by inhibiting HDAC activities the progression of hearing loss can be modulated. 

From preclinical studies to clinical trials, HDAC inhibitors have demonstrated powerful therapeutic effects in various cancers. HDAC inhibitors can significantly attenuate tumor burden by limiting tumor growth and restraining aberrantly proliferated vessels. HDAC inhibitors can also induce DNA damage, cell cycle arrest, apoptosis, and autophagy to promote cancer cell death mentioned above [28]. Indeed, Merkel cell carcinoma (MCC) is partially determined by histone post-translational modifications, including histone acetylation, methylation, and phosphorylation. This malignant behavior of MCC cells can be reverted with HDAC inhibitors [29].

The most striking observation in this study was that VPA inhibited the KCNQ variant-induced HL. VPA is known to be an HDAC inhibitor [30], which selectively inhibits the catalytic activity of class I and class IIa HDACs [31]. Above all, in class I and class IIa HDACs, only HDAC1 contributed to reducing KCNQ4 and HSP90β expressions (Appendix A). Previous investigations show that VPA targets HDAC1 and HDAC2 [32]. However, our study showed that only HDAC1 had an effect on the expressions of KCNQ4 and HSP90β. Forthun et al. [33] reported that HSP90β is upregulated in response to VPA and increased survival in more than 20% of patients. In addition, our study showed that the upregulation of KCNQ4 and HSP90β by VPA (Figure 4A,B) increased the survival of the OHCs (Figure 3A,B). The overexpression of HDAC1 contributed to the downregulation of the expressions of KCNQ4 and HSP90β (Figure 4C–E). This observation provides an indication for the potential therapeutic application of VPA to augment the HSP90β-KCNQ4/HDAC1 signaling cascade. Moreover, VPA markedly increased the physical interaction between KCNQ4 and HSP90β (Figure 5A), which caused the upregulation of the protein expression of both KCNQ4 and HSP90β in HEK 293T cells. Furthermore, HDAC1 disrupted the KCNQ4–HSP90β physical interaction (Figure 5B), which promoted the reduction in their protein expressions. In addition, CHIP (C-terminal of HSP70-interacting protein, NM_005861.2), major E3 ubiquitin ligase for HSP90 client proteins, associate with HSP70-CHIP complexes and to be targeted for degradation via ubiquitination-proteasome pathway. This pathway controls KCNQ4 homeostasis via the HSP40-HSP70-HOP-HSP90 chaperone pathway and the ubiquitin-proteasome pathway [19]. The high potential of the ubiquitin-proteasome system in regulating many human diseases is beginning to receive a broad recognition. Proteins of the ubiquitin-proteasome system and E3 ubiquitin ligases, in particular, are emerging as promising molecular targets for drug discovery in various diseases, including autoimmune and neurodegenerative [34].

In the present study, the reduction of HDAC1 activity by VPA increased KCNQ4 expression and protected against HL in the cochlea. In addition, we found that VPA non-treated KCNQ4 p.W276S variants showed significant OHC death in the middle and base regions of the cochlea. However, treatment with VPA promoted an increased survival of these OHCs in the cochlea of the KCNQ4 p.W276S variants (Figure 3A,B). In addition, Wakizono et al. [35] reported that VPA, along with growth factors (EGF and bFGF) combination treatment, recovers spiral ganglion neurons. These results also provided beneficial outcomes for hearing in both the click and tone burst evaluations (Figure 1B,C).

There are some studies reporting an ototoxic effect of VPA [36,37]. This contradicts our current study’s findings. Several factors may account for this discrepancy. The effect of VPA might differ according to the dose, age of the recipient, combination with other drugs, frequency of administration, and duration of treatment. VPA might also act differently depending on the type of hearing loss. In genetic hearing loss, such as in our study, the effect of gene regulation may be more effective than the general ototoxic effect. In addition, combinatorial CHIR99021 and VPA treatment is in the clinical trials (FX-322) to treat sensorineural hearing loss. Precise application of VPA on different hearing condition and further patho-mechanistic studies are needed.

The reports showed that heterozygous KCNQ4 dimer conformation was attenuated with the KCNQ4 WT and decreased the surface expression of the KCNQ4 p.W276S variant with the WT co-expression [38]. Furthermore, HSP90β significantly improved the cell surface expression of the KCNQ4 WT and KCNQ4 variants and promoted the HSP90β rescue of the KCNQ4 variant channel function, as a result of the surface KCNQ4 expression being significantly improved by the HSP90β molecular chaperone. Moreover, the KCNQ4 variant protein expression was rescued by the dose-dependent expression of HSP90β. Functional KCNQ channels are assembled in homo- or heterotetrameric pore-forming subunits. Therefore, we proposed that the KCNQ4 variant-induced HL is a result of an attenuated KCNQ4 pore assembling and that rescue KCNQ4 pore-forming subunits occur by the molecular chaperone HSP90β. 

Gene therapy that replaces the missing gene or corrects the defect site is developing progressively these days, while gene therapy has been attempted in genetic hearing loss as well. For example, in delivering Vglut3 by using an adeno-associated virus 1 to Vglut3 knockout mouse, the Vglut3 expression was restored in the inner hair cell, which improved functional hearing [39]. Similarly, injecting the Cas9-guide RNA complex into the Tmc1 Beethoven mouse model, corrected the single defective base pair and rescued the hearing [40]. Moreover, gene editing has previously been conducted in a KCNQ4 p.W276S mouse model. Kv7.4 channel activity and functional hearing were restored by dual adeno-associated virus (AAV) system using CRISPR-based gene therapy [41]. These are all promising strategies for genetic hearing loss. However, one key point in these gene therapies remains, whereby they were performed in the mouse pup, which is still in its developmental stage. In humans, the representative time period is before birth, and performing an intra-cochlear gene delivery through the uterus will be extremely challenging [42]. Delaying the hearing loss progression has a great meaning to meeting the practical time for gene therapy. Our study showed that the progression of hearing deterioration could be inhibited by VPA in a KCNQ4 p.W276S variant model.

In summary, VPA suppressed HDAC1, leading to upregulation of KCNQ4, HSP90β expression, and interaction between KCNQ4 and HSP90β. Further study is needed to understand the regulation of downstream target gene by inhibiting HDAC1. Furthermore, although we observed that inhibiting HDAC1 recovers KCNQ4 and HSP90β expression, it remains to be elucidated whether this can restore potassium channel current. For this reason, HDAC inhibitor application maybe needed in the future. Despite the search for insufficient HDAC1 downstream gene regulation, VPA is an excellent candidate drug for inhibiting KCNQ4 variant-induced genetic hearing loss.

## 4. Materials and Methods

### 4.1. VPA Treatment and Generation of Hearing Loss Animals

We used KCNQ4 p.W276S variant mice with a C57BL/6N background, which were aged from 3 weeks to 16 weeks. VPA was administered by daily intraperitoneal injection (200 mg/kg) or subcutaneously implanted with an osmotic pump (200 mg/kg/day) for 4 weeks. The Alzet^®^ micro-osmotic pump was purchased from DURECT (DURECT Corporation, Cupertino, CA, USA). The mice were sheltered in a standard-conditioned vivarium, with free access to food and water. Cages were changed every week, and food and water were replenished every three days. Mice were monitored daily for the health status and any signs of discomfort. All mice were healthy until sacrifice. The care and use of the animals in this study were approved by the Institutional Animal Care and Use Committee at Chonnam National University Medical School (CNUHIACUC-21045). KCNQ4 variant mice were kindly provided by professors Jinsei Jung and Jae Young Choi (Yonsei University, College of Medicine) [7,41]. 

### 4.2. Auditory Brainstem Response for an Animal Hearing Evaluation

We recorded the ABR with a 3RZ6 TDT system (Tucker-Davis Technologies, Alachua, FL 32615, USA), which provided stimuli ranging from clicks to tone bursts. Needle electrodes of 1.5 mm in length were inserted sub-dermally at the dorsal midline between the eyes (none inverting), at the scalp, and posterior to both pinnae. At each frequency, we tested various stimuli intensity levels in decreasing order, from 90 to 20 dB of the visual ABR threshold. Mice were anesthetized using a cocktail of ketamine 80 mg/kg and xylazine 10 mg/kg while performing ABR and remained asleep during all ABR recordings.

### 4.3. Expression Constructs

pcDNA3-HA-HSP90β (plasmid #22487) and pcDNA3.1-KCNQ4-EGFP-8xHis expression vectors were purchased from Addgene (plasmid #111453, Watertown, MA, USA). pCs2+-3myc-HDAC1, pCs2+-3myc-HDAC2, pcDNA3.1-HDAC3-Flag, and pAP3neo-HDAC4-Flag were kindly provided by Professor. Gwang Hyeon Eom (Chonnam National University Medical School, Hwasun, Korea).

### 4.4. Immunohistochemistry for OHCs

Mice were euthanized with a cocktail of ketamine and xylazine (80 and 10 mg/kg, respectively) before the extraction of the cochlea. Using a 0.5-cc syringe, a hole was created at the apex of each cochlea and the cochlea was then perfused with phosphate-buffered solution (PBS), followed by 4% paraformaldehyde (PFA). It was then immersed in a 4% PFA solution for 1 h with gentle rotation at 4 °C. The cochlea were rinsed twice with PBS before decalcification with 0.12 mM ethylenediaminetetraacetic acid (EDTA) for 1 h on gentle rotation at 4 °C. To reveal the Organ of Corti, the bone and stria vascularis surrounding the cochlea were dissected, and the tectorial membrane was removed. Three small pieces were cut from each cochlea (apex, middle, and base), and the tissue samples were immersed in a blocking buffer for 1 h at room temperature (RT) before being incubated with primary antibodies overnight at 4 °C. After three washing cycles with 0.1% PBS-T (30 min each wash), the samples were incubated in secondary antibodies for 2 h at room temperature. Finally, the samples were washed three times with 0.1% PBS-T for 30 min, stained for 3 min with DAPI and washed in PBS for 30 min. A vector protection solution was used to mount the samples on glass slides and the slides were examined using an LSM 800 laser scanning microscope (Carl Zeiss Microscopy GmbH, Promenade 10, 07745 Jena, Germany). The following antibodies and titers were utilized: myosin-7a (1:200, # 25-6791, Proteus), Prestin (1:1000, # A12379, Cell Signaling Technology, Danvers, MA 01923, USA), KCNQ4 (1:200, #PA5-101767, Thermo Fisher Scientific Inc., Waltham, MA, USA) and DAPI (1:10000, Invitrogen, Carlsbad, CA 92008, USA).

### 4.5. Counting of Hair Cells

The number of OHCs in the cochlea was ascertained. The cochlea were divided into the apical, middle, and basal turns, and the hair cells in each turn were counted under 200× magnification. The number of hair cells per 100 µm cochlear turn length was averaged for each group (*n* = 5). 

The lengths of the cochlear turns were measured for each study group. Confocal z-stacks of three areas were generated from each cochlea using a high-resolution confocal microscope (LSM 800 laser scanning microscope). Image stacks were converted using image editing software. At least 12 hair cells were found in each cochlear turn (apex, middle, and base).

### 4.6. Antibodies

Primary antibodies used in this study were anti-histone H4ac (39926, Active Motif, Carlsbad, CA 92008, USA), anti-Myo7a (#25-6790, Proteus, Ramona, CA 92065, USA), anti-Prestin (MBS423494, MyBioSource, Inc., San Diego, CA 92195, USA), anti-Flag (F3165), anti-Actin (A2066, Sigma-Aldrich, Inc., St. Louis, MO, USA), anti-KCNQ4 (#PA5-101767, Thermo Fisher Scientific Inc., Waltham, MA, USA), anti-HSP90 beta (ab32568, Abcam, Waltham, MA, USA), anti-HA (#3724), and anti-Myc (#2276, Cell Signaling, Beverly, MA, USA). Anti-GFP (sc-9996) and anti-His (sc-8036) antibodies were purchased from Santa Cruz Biotechnology, Inc. (Santa Cruz, CA, USA). (HRP)-conjugated secondary antibodies were from Thermo Fisher Scientific Inc., (Waltham, MA, USA), including goat anti-mouse-HRP (31430), and goat anti-rabbit-HRP (32460).

### 4.7. Protein Preparation

For Western blot and immunoprecipitation, cell lysates were obtained using NP lysis buffer (1% Nonidet-P40, 50 mM Tris pH 8.0, 150 mM NaCl, 10 mM NaF, 1 mM Na_3_VO_4_, 5 mM EDTA, 1 mM EGTA, 1 mM PMSF, 1 mM DTT) supplemented with protease inhibitor cocktail.

### 4.8. Co-Immunoprecipitation 

The transfected cells were lysed in NP40 lysis buffer supplemented with protease inhibitor cocktail (P8340, Sigma-Aldrich, St. Louis, MO, USA), on ice for 30 min. Cell lysates were cleared by centrifugation at 14,000 rpm for 10 min at 4 °C and incubated with primary antibodies, as indicated, at 4 °C for 24 h. The protein complexes were isolated and purified using protein A/G PLUS-Agarose (sc-2003, Santa Cruz), following the manufacturer’s protocol, and analyzed by Western blot.

### 4.9. In Vitro HEK293T Culture and Transfection

HEK293T cells were used for all experiments. These cells were maintained according to the manufacturer’s instructions. All transfections were performed using TurboFect™, as described by the manufacturer (Thermo Scientific). Following transfection, the cells transfected with plasmid DNA were incubated at 37 °C for 24 h.

### 4.10. RNA Isolation and Real-Time Polymerase Chain Reaction

RNA isolation and real-time polymerase chain reaction were performed for downstream gene analysis. Cochlear whole tissues were harvested, and total RNA was extracted with Trizol reagent (Invitrogen, Carlsbad, CA 92008, USA). The quantity of RNA was determined by spectrophotometry (Spectrophotometer ND-1000 Nanodrop, Technologies Inc., Wilmington, MA, USA) at an absorbance of A260/A280 nm. The results were analyzed using ND-1000 Software. The experiments were performed three times, and each sample was assayed in triplicate. Denaturation was performed for 10 min and 10 s at 95 °C, and the annealing phase took place at 62 °C for 20 s, followed by 72 °C for 30 s with 40 cycles. The primers for each gene are as follows: GAPDH_For (5′-ACC ACA GTCCAT GCC ATC AC-3′); GAPDH_Rev (5′-TCCACCACCCTG TTG CTG TA-3′); SMN_For (5′-GAATGCCACAACTCCCTTG-3′); SMN_Rev (5′-GCAGCCGTCTTCTGACCAA-3′).

### 4.11. Statistical Analysis

All data are presented as mean ± SEM. Differences among groups were assessed by a one-way analysis of variance (ANOVA) followed by a post hoc Tukey’s test. Comparisons between the two groups were performed using a Student’s *t*-test. All statistical analyses were performed with GraphPad Prism 6.0. A value of *p* < 0.05 was considered statistically significant.

## 5. Conclusions

In this study, we found that VPA inhibited HL progression in a KCNQ4 p.W276S variant model. VPA suppressed HDAC1, leading to upregulation of KCNQ4, HSP90β expression, and interaction between KCNQ4 and HSP90β. VPA is a candidate drug that slows down HL progression to increase the time window for the definite treatment of KCNQ4 p.W276S variant that induces genetic HL.

## Figures and Tables

**Figure 1 ijms-24-05695-f001:**
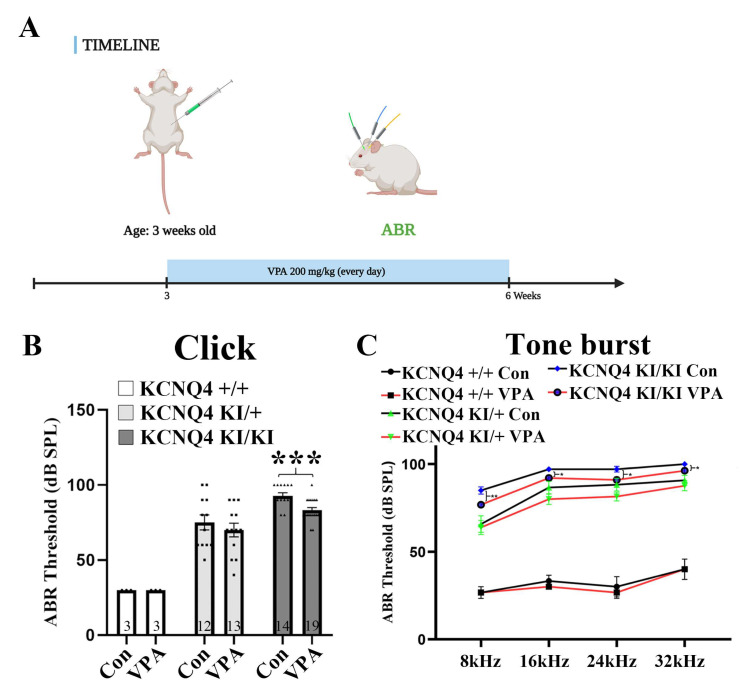
Systemic injection of VPA reduces hearing loss in the KCNQ4 p.W276S variant. (**A**) Experimental timeline. VPA was injected intraperitoneally into the WT, hetero KCNQ4 p.W276S variant, and homo KCNQ4 p.W276S variant mice. Three weeks later, the auditory brainstem response (ABR) was evaluated using click and tone burst stimuli. (**B**) VPA attenuated KCNQ4 p.W276S variant-induced hearing loss as shown by ABR click results. *** *p* < 0.001, numerals in bar graphs are the numbers of samples. Error bars represent S.E.M (standard error of the mean). Con: control; VPA: valproic acid. (**C**) The tone burst results showed a significantly decreased at 8, 16, 24, and 32 kHz in VPA homo KCNQ4 treated mice. * *p* < 0.05, ** *p* < 0.01, error bars represent S.E.M.

**Figure 2 ijms-24-05695-f002:**
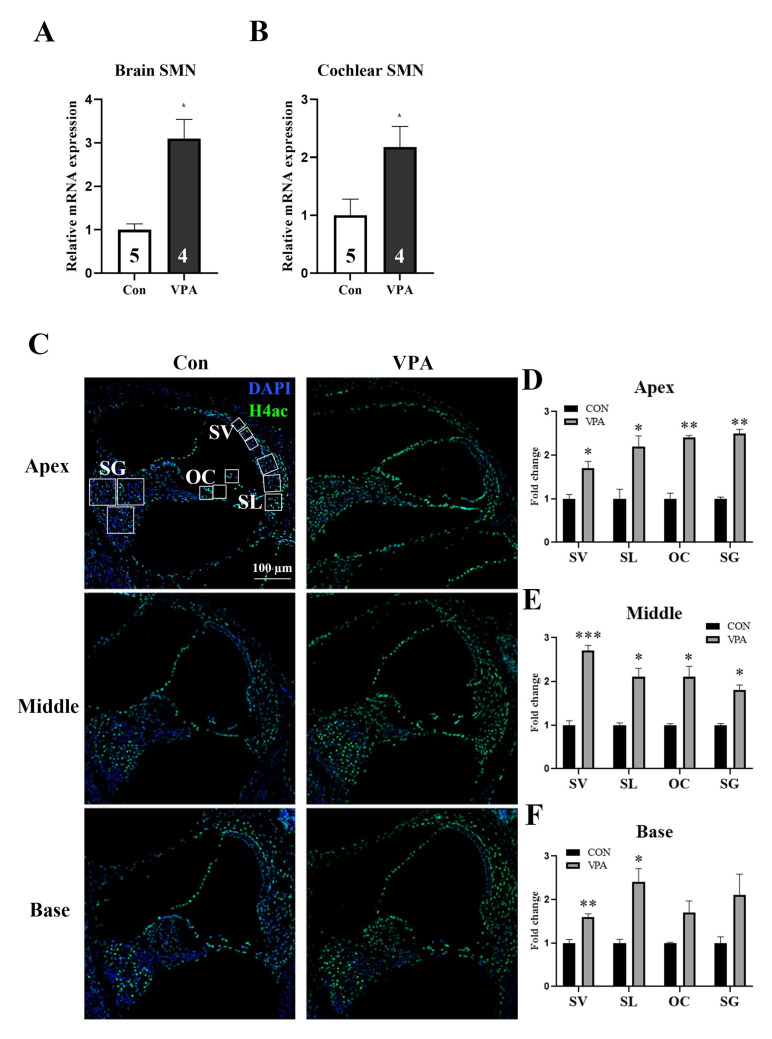
VPA increases SMN and histone H4 acetylation in the cochlea. (**A**) VPA treatment increased SMN mRNA levels in the brain. * *p* < 0.05, numerals in bar graphs are the numbers of samples. Error bars represent S.E.M. Con: control; VPA: valproic acid. (**B**) VPA treatment increased SMN mRNA levels in the cochlea. * *p* < 0.05, numerals in bar graphs are the numbers of samples. Error bars represent S.E.M. (**C**) VPA treatment enhanced histone H4 acetylation in the cochlear apex, middle, and base regions. Representative image of control mice in apex region. (SV: stria vascularis, SL: spiral ligament, OC: Organ of Corti, SG: spiral ganglion). (**D**–**F**) Quantification result of mice cochlea immunostaining. Whole-mount immunostaining of the cochlea with an antibody directed towards acetyl-histone H4 (H4ac, green) with DAPI. Cochlear apex, middle, and base H4ac intensity were calculated in the SV, SL, OC, and SG regions, each selected from the same three square areas. * *p* < 0.05, ** *p* < 0.01, *** *p* < 0.001. Error bars represent S.E.M.

**Figure 3 ijms-24-05695-f003:**
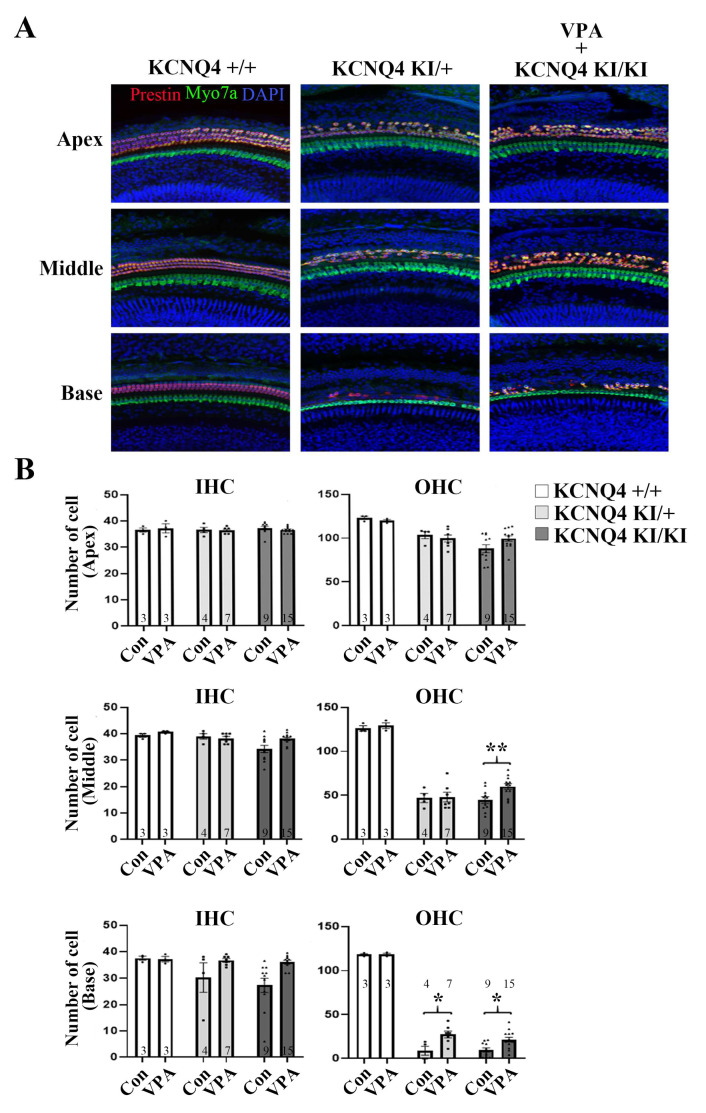
VPA protected-OHCs from cell death in the KCNQ4 p.W276S variant. (**A**) Representative image of mice cochlea immunostaining. Whole-mount immunostaining of the cochlea with antibodies directed towards Myo7a (green), as a marker for IHCs and OHCs, and Prestin (red) to show the OHCs. Cochlear apex, middle, and base outer hair cells were counted in homo mice and were significant at the base turn of VPA-treated homo compared to VPA non-treated homo mice. (**B**) Quantification results. * *p* < 0.05, ** *p* < 0.01, numerals in bar graphs are the numbers of samples. Error bars represent S.E.M.

**Figure 4 ijms-24-05695-f004:**
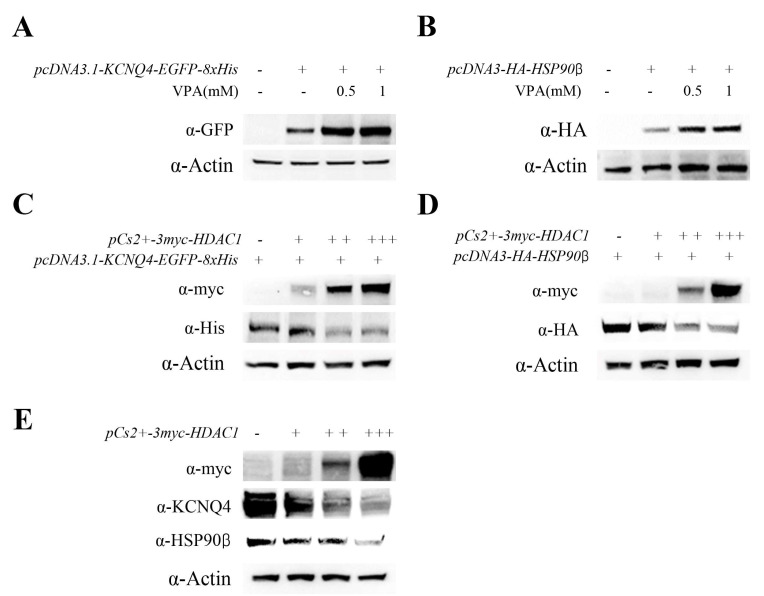
VPA increased KCNQ4 and HSP90β protein expression by inhibiting HDAC1. (**A**) VPA increased transient transfection of *pcDNA3.1*–*KCNQ4*–*EGFP*–*8xHis* protein expression in HEK293T cells. (**B**) Transient transfection of *pcDNA3*–*HA*–*HSP90β* increased protein expression in HEK293T cells treated with VPA. (**C**) Transient transfection of *pcDNA3.1*–*KCNQ4*–*EGFP*–*8xHis* decreased by *pCs2+*–*3myc*–*HDAC1* dose-dependent expression in HEK293T cells. (**D**) Transient transfection of *pcDNA3*–*HA*–*HSP90β* decreased by *pCs2+*–*3myc*–*HDAC1* dose-dependent expression in HEK293T cells. (**E**) Endogenous KCNQ4 and HSP90β reduced protein expression by dosage-dependent transient transfection of *pcDNA3*–*HA*–*HSP90β* in HEK293T cells.

**Figure 5 ijms-24-05695-f005:**
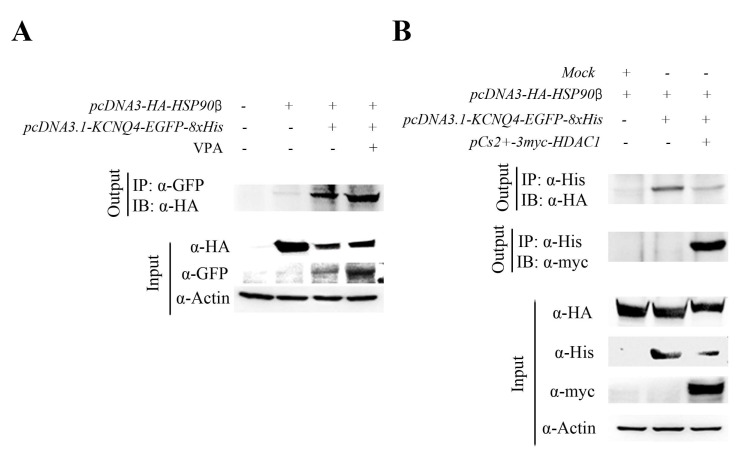
VPA increased HSP90β–KCNQ4 interaction by inhibiting KCNQ4 direct interruption with HDAC1. (**A**) Immunoprecipitation results show the physical interaction increase by VPA between exogenous KCNQ4 and HSP90β in HEK293T cells. (**B**) HDAC1 interrupted protein interaction KCNQ4 with HSP90β by directly binding with KCNQ4.

**Figure 6 ijms-24-05695-f006:**
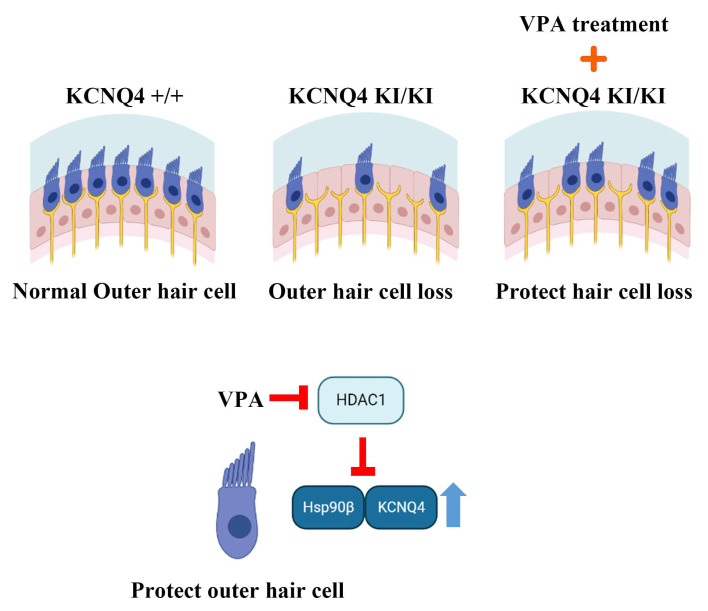
Schematic Diagram of VPA protecting against KCNQ4 variant-induced hair cell loss. KCNQ4 p.W276S variant induced outer hair cell loss, which was protected by a VPA treatment increase in HSP90β KCNQ4 binding, with expression through the inhibition of HDAC1 activation.

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
