# Peer review of "Valproic Acid Inhibits Progressive Hereditary Hearing Loss in a KCNQ4 Variant Model through HDAC1 Suppression"

_ijms, 2023, doi:10.3390/ijms24065695_

Round 1

Reviewer 1 Report

Nam et al. studied the role of Valproic acid to treat Hearing loss, This is well-written and concise, but some factors need to discuss which are mentioned here

1. While treating hearing loss with Valproic acid, does it affects the functioning of other organs?

2. Some studies have shown that Valproic acid has an ototoxic effect on the human body (for example; https://doi.org/10.2169/internalmedicine.42.1153and DOI: https://doi.org/10.1212/WNL.40.12.1896). how this aspect will be justified in this study? 

3. The role of Valproic acid in treating hearing loss has been recently reported (PMID: 34806649), and it also needs to be added to the discussion.

4. Is there any limitation of this study? if yes please mention it in the discussion.

Author Response

Thank you for giving us the opportunity to submit a revised draft of our manuscript titled “Valproic acid inhibits progressive hereditary hearing loss in a KCNQ4 variant model through HDAC1 suppression” to IJMS. We truly appreciate the time and effort of the reviewers for providing their valuable feedback on our manuscript. We have been able to incorporate relevant changes to reflect most of the suggestions provided by the reviewers. The changes have been highlighted within the manuscript. Here is a point-by-point response to the reviewers’ comments and concerns.

Comment:

  1. While treating hearing loss with Valproic acid, does it affects the functioning of other organs?

Response:

We thank the reviewer for their helpful review on our manuscript. Many possible side effects of valproic acid had been reported including hepatotoxicity However, most of the studies that reported the side effect used much higher dose (i.e. 1000mg/kg) than what was used in our study (200mg/kg). We checked the condition of the mouse very carefully every day. All mice were healthy until sacrifice. These are now mentioned in the manuscript material & methods.

Cages were changed every week, and food & water were replenished every three days. Mice were daily monitored for the health status and any signs of discomfort. All mice were healthy until sacrifice. (lines 331 to333).

Comment:

  1. Some studies have shown that Valproic acid has an ototoxic effect on the human body (for example; https://doi.org/10.2169/internalmedicine.42.1153 and DOI: https://doi.org/10.1212/WNL.40.12.1896). how this aspect will be justified in this study?

Response:

We appreciate the reviewer's thorough and insightful comment. We agree that there is a discrepancy between those reports and our study. In case of those human studies, basically valproic acid was used in higher dose than 200mg/kg/day which was used in our experiment. Tinnitus was one of side effect of high dose valproic acid. In addition, instead of general ototoxicity our study was conducted in a genetic hearing loss model (hearing loss caused by KCNQ4 variant). This genetic model shows a reduction in potassium channel activity, which was restored by appropriate dose of valproic acid. As the reviewer mentioned in the comment below, there are other studies using valproic acid for hearing loss treatment. A clinical trial, FX-322 which contains valproic acid, is applied to treat sensorineural hearing loss in human. 

We have included this matter in lines (286-294) in the revised manuscript

There are some studies reporting an ototoxic effect of VPA [36,37]. This contradicts our current study's findings. Several factors may account for this discrepancy. The effect of VPA might differ according to the dose, age of the recipient, combination with other drugs, frequency of administration, and duration of treatment. VPA might also act differently depending on the type of hearing loss. In genetic hearing loss, such as in our study, the effect of gene regulation may be more effective than the general ototoxic effect. In addition, combinatorial CHIR99021 and VPA treatment is in the clinical trials (FX-322) to treat sensorineural hearing loss. Precise application of VPA on different hearing condition and further patho-mechanistic studies are needed. (lines 286 to 294).

We have also included FX-322 trials in lines (61-65) in the revised manuscript

In addition, the combination of CHIR99021 and VPA is in the clinical trials (FX-322) to treat sensorineural hearing loss [15]. HDAC inhibitors, which are central players in epi-genetic gene modification and regulation of intracellular signaling, are involved in HL gene regulation [11]. Therefore, VPA treatment is also well-suited for modulating HL in a KCNQ4 variant model. (lines 61 to 65).

Comment:

  1. The role of Valproic acid in treating hearing loss has been recently reported (PMID: 34806649), and it also needs to be added to the discussion.

Response:

We are thankful to the reviewer for the useful suggestions. We have included suggest report in lines (282-284) in the revised manuscript.

In addition, Wakizono et al. [35] reported that VPA, along with growth factors (EGF and bFGF) combination treatment, recovers spiral ganglion neurons. (lines 282 to 284).

Comment:

  1. Is there any limitation of this study? if yes please mention it in the discussion.

Response:

We appreciate the reviewer's thoughtful recommendations. We have not confirmed the effect of VPA on downstream target genes and potassium channel current .

We have included in lines (322-329) in the revised manuscript.

In summary, VPA suppressed HDAC1, leading to upregulation of KCNQ4, HSP90β expression, and interaction between KCNQ4-HSP90β. Further study is needed to under-stand the regulation of downstream target gene by inhibiting HDAC1. Furthermore, although we observed that inhibiting HDAC1 recovers KCNQ4 and HSP90β expression, it remains to be elucidated whether this can restore potassium channel current. For this reason, HDAC inhibitor application maybe needed in the future. Despite the search for insufficient HDAC1 downstream gene regulation, VPA is an excellent candidate drug for inhibiting KCNQ4 variant-induced genetic hearing loss. (lines 322 to 329).

Reviewer 2 Report

Yoon Seok Nam and co-authors present a quality and well-written experimental manuscript focused on valproic acid inhibits progressive hereditary hearing loss in a KCNQ4 variant model through HDAC1 suppression

Authors investigated the protective effects of VPA on the auditory functions of the KCNQ4 variant and analyzed the feasibility of using an HDAC inhibitor to modulate late-onset genetic HL.

Authors demostrated that systemic injections of VPA attenuated hearing loss and protected the cochlear hair cells from cell death in the KCNQ4 p.W276S mouse model. VPA activated its known downstream target, survival motor neuron gene, and increased acetylation of histone H4 in the cochlea, demonstrating that VPA treatment directly affects the cochlea. In addition, treatment with VPA increased the KCNQ4 binding with HSP90β by inhibiting HDAC1 activation in HEI-OC1 in an in vitro study. VPA is a candidate drug for inhibiting late-onset progressive hereditary hearing loss from the KCNQ4 p.W276S variant.

Finally, authors conclude that the progression of hearing deterioration could be inhibited by VPA in a KCNQ4 p.W276S variant model. They also conclude that VPA is an excellent candidate drug for inhibiting KCNQ4 variant-induced genetic hearing loss.

==============================

Overall, the manuscript is highly valuable for the scientific community and should be accepted for publication after the corrections are made.

Other comments:

1) Please check for typos throughout the manuscript.

2) Section 2.5. With regards to Ubiquitin-proteasome pathway – authors are kindly encouraged to cite the following article that discussed various aspects of protein homeostasis relevant for KCNQ4. DOI: 10.1007/s12668-016-0233-x

Author Response

Thank you for giving us the opportunity to submit a revised draft of our manuscript titled “Valproic acid inhibits progressive hereditary hearing loss in a KCNQ4 variant model through HDAC1 suppression” to IJMS. We truly appreciate the time and effort of the reviewers for providing their valuable feedback on our manuscript. We have been able to incorporate relevant changes to reflect most of the suggestions provided by the reviewers. The changes have been highlighted within the manuscript. Here is a point-by-point response to the reviewers’ comments and concerns.

Comment:

  1. Please check for typos throughout the manuscript

Response:

We thank the reviewer for their helpful review of our manuscript. We have carefully checked the manuscript and corrected all typos.

abcabc. (lines 301 to 303).

Comment:

  1. Section 2.5. With regards to Ubiquitin-proteasome pathway – authors are kindly encouraged to cite the following article that discussed various aspects of protein homeostasis relevant for KCNQ4. DOI: 10.1007/s12668-016-0233-x

Response:

We are thankful to the reviewer for the important observations and useful suggestions. We have included suggest report in lines (269-277) in the revised manuscript.

In addition to, CHIP (C-terminal of HSP70-interacting protein, NM_005861.2), major E3 ubiquitin ligase for HSP90 client proteins, associate with HSP70-CHIP complexes and to be targeted for degradation via ubiquitination-proteasome pathway. This pathway controls KCNQ4 homeostasis via the HSP40-HSP70-HOP-HSP90 chaperone pathway and the ubiquitin-proteasome pathway [19]. The high potential of the ubiquitin-proteasome system in regulating many human diseases is beginning to receive a broad recognition. Proteins of the ubiquitin-proteasome system and E3 ubiquitin ligases, in particular, are emerging as promising molecular targets for drug discovery in various diseases, including autoimmune and neurodegenerative [34]. (lines 269 to 277).

Reviewer 3 Report

Research article on progressive hereditary hearing loss therapy from Dr. Nam and colleagues.

In this interesting in vivo-based study the authors evaluated whether using Valproic acid (VPA), which is an important and commonly used histone deacetylase (HDAC) inhibitor for class I (HDAC1, 2, 3, and 8) and class IIa (HDAC4, 5, 7, and 9), might attenuate hearing loss and protected the cochlear hair cells from cell death in the KCNQ4 p.W276S mouse model. Results indicate that VPA is a candidate drug for inhibiting late-onset progressive hereditary hearing loss. Due to the lack of figures, a proper revision of the manuscript is impossible. The decision is therefore reconsider after major revision

1.       The main comment is that, unfortunately, figures were not available for the revision so it was impossible to do it properly. Figures should be uploaded with the Revised version of the manuscript

2.       Please improve the rationale for the selection of HDAC inhibitors for hearing loss therapy in the introduction

1.       Authors are kindly encouraged to include these recently published and detailed reviews on HDAC inhibitors and cancer (PMID: 35350569 and PMID: 33117804).

2.       Methods sections from 4.2 to 4.11 should be improved with supporting references

3.       Study lmitations should be included before conclusions.

Author Response

Thank you for giving us the opportunity to submit a revised draft of our manuscript titled “Valproic acid inhibits progressive hereditary hearing loss in a KCNQ4 variant model through HDAC1 suppression” to IJMS. We truly appreciate the time and effort of the reviewers for providing their valuable feedback on our manuscript. We have been able to incorporate relevant changes to reflect most of the suggestions provided by the reviewers. The changes have been highlighted within the manuscript. Here is a point-by-point response to the reviewers’ comments and concerns.

Comment:

  1. The main comment is that, unfortunately, figures were not available for the revision so it was impossible to do it properly. Figures should be uploaded with the Revised version of the manuscript

Response:

We thank the reviewer for their helpful review of our manuscript. We uploaded all figures in the MDPI style for the revised manuscript.

Comment:

  1. Please improve the rationale for the selection of HDAC inhibitors for hearing loss therapy in the introduction

Response:

We are thankful to the reviewer for the important observations and useful suggestions. We have included several reports suggesting the therapeutic effect of HDAC inhibitors and valproic acid for hearing preservation in lines (53-65) of the revised manuscript.

HDAC inhibitors have been used for a wide variety of purposes including anticancer, anti-aging, anti-inflammatory, antioxidative, and neuroprotection [11]. Similarly, the ability of HDAC inhibitors, SAHA and trichostatin A, to manage hearing deficits including drug-induced- or noise-induced HL has been investigated [11]. Treatment with sodium butyrate showed a protective effect on gentamicin-induced hair cell loss through HDAC1 modulation [12]. Valproic acid (VPA), originally an anticonvulsant, also provided an HDAC inhibitory effect and regulated HDAC class I and II [13]. Interestingly, VPA showed its antiepileptic effect by preserving the KCNQ family M-current activity [14]. In addition, the combination of CHIR99021 and valproic acid is in the clinical trials (FX-322) to treat sensorineural hearing loss [15]. HDAC inhibitors, which are central players in epigenetic gene modification and regulation of intracellular signaling, are involved in HL gene regulation [11]. Therefore, VPA treatment is also well-suited for modulating HL in a KCNQ4 variant model. (lines 53 to 65).

Comment:

  1. Authors are kindly encouraged to include these recently published and detailed reviews on HDAC inhibitors and cancer (PMID: 35350569 and PMID: 33117804).

Response:

We have included suggest report in lines (245-253) in the revised manuscript.

From preclinical studies to clinical trials, HDAC inhibitors have demonstrated powerful therapeutic effects in various cancers. HDAC inhibitors can significantly attenuate tumor burden by limiting tumor growth and restraining aberrantly proliferated vessels. HDAC inhibitors can also induce DNA damage, cell cycle arrest, apoptosis, and autophagy to promote cancer cell death mentioned above [28]. Indeed, Merkel cell carcinoma (MCC) is partially determined by histone post-translational modifications, including histone acetylation, methylation, and phosphorylation. This malignant behavior of MCC cells can be reverted with HDAC inhibitors [29].

Comment:

  1. Methods sections from 4.2 to 4.11 should be improved with supporting references.

Response:

We thank the reviewer for their helpful review of our manuscript.

We have included lines (333, 337-339, 376-377) in the revised manuscript.

We used KCNQ4 p.W276S variant mice with a C57BL/6N background, which were aged from 3 weeks to 16 weeks. (lines 332 to 333)

Cages were changed every week, and food & water were replenished every three days. Mice were daily monitored for the health status and any signs of discomfort. All mice were healthy until sacrifice. (lines 337 to 339).

KCNQ4 (1:200, #PA5-101767, Thermo Fisher Scientific Inc., Waltham, MA, USA). (lines 376 to 377).

Comment:

  1. Study limitations should be included before conclusions.

Response:

We appreciate the reviewer's thoughtful recommendations. We did not confirm the downstream target gene expression and potassium channel current By VPA treatment. We have included in lines (322-329) in the revised manuscript.

In summary, VPA suppressed HDAC1, leading to upregulation of KCNQ4, HSP90β expression, and interaction between KCNQ4-HSP90β. Further study is needed to understand the regulation of downstream target gene by inhibiting HDAC1. Furthermore, although we observed that inhibiting HDAC1 recovers KCNQ4 and HSP90β expression, it remains to be elucidated whether this can restore potassium channel current. For this reason, HDAC inhibitor application maybe needed in the future. Despite the search for insufficient HDAC1 downstream gene regulation, VPA is an excellent candidate drug for inhibiting KCNQ4 variant-induced genetic hearing loss. (lines 322 to 329).

Round 2

Reviewer 3 Report

The manuscript can be accepted in the present form